# OpenReview forum: "Let me PASS: Formalization Driven Prompt Jailbreaking via Reinforcement Learning"
_ICLR.cc/2026/Conference — ICLR 2026 Conference Withdrawn Submission_

### Official Review · Reviewer_SQNF · 2025-10-14

**Soundness:** 2
**Presentation:** 2
**Contribution:** 3
**Rating:** 4
**Confidence:** 5

**Summary:**

The paper introduces PASS, a framework for constructing stealthy and adaptive prompt jailbreak attacks on aligned large language models (LLMs). Instead of relying on static, template-based attacks, PASS uses reinforcement learning to iteratively transform malicious prompts into formalized, decomposed representations. It further leverages a GraphRAG module to extract and reuse formalized knowledge from successful attacks. The framework is evaluated on several open-source and proprietary LLMs and demonstrates significantly higher attack success rates compared to existing baseline and recent attack methods.

**Strengths:**

- **Strong quantitative gains:** Results in Table 1 (Page 8) show that PASS consistently achieves higher attack success rates (e.g., 99.03% on DeepSeek-V3) than all competing methods across two benchmarks and four LLMs, including the more robust Claude Sonnet 4.

- **Multi-step, compositional design:** As shown in Figure 2, the method enables dynamic chaining of symbolic abstraction, logical encoding, and domain specialization, supporting compositional prompt obfuscation and improved flexibility.

- **Sophisticated RL framework:** The methodology is clearly formulated, with precise RL optimization equations (Page 7), using PPO with Generalized Advantage Estimation and a reward decomposition that balances attack success, stealth, and fidelity.

- **New jailbreak method:** They propose a novel jailbreak attack method, named PASS, based on the formalization of jailbreak prompts. Our method employs reinforcement learning to achieve multi-round jailbreaking.

**Weaknesses:**

- Incorporate recent state-of-the-art attack baselines (e.g., AutoDAN-Turbo[3], BOOST + GPTFuzzer[1], TAP[2]) to strengthen the comparative analysis; please add 1–2 newer attack methods.

- **Insufficient ablation on the formalization action space:** Section 4 and Figure 3 enumerate symbolic, logical, mathematical, etc., but Table 1 does not isolate each action’s contribution—no ablation or variant study shows which actions drive stealthiness or transferability.

- The BOOST paper, which discusses model-safety alignment blind spots, is not cited.

- Relying on an auxiliary LLM to compute reward risks overfitting or bias (e.g., a permissive or misaligned judge), raising concerns about cross-model generalizability.

- **No formal robustness or detection evaluation:** despite high attack rates, there is no experiment or discussion of whether existing or new defenses (e.g., “Defending Jailbreak Prompts via In-Context Adversarial Game” or “Adversarial Tuning”) can counter or adapt to PASS, leaving real-world durability untested.

**Questions:**

1. Why didn't you directly use the strongReject score together with ASR as evaluation metrics?
Combining them would jointly assess rejection robustness and transcription accuracy, so please clarify why they were omitted.
2. How robust is PASS to updated alignment defenses utilizing in-context adversarial games or prompt adversarial tuning? Can the authors provide empirical results against such dynamic defense strategies or at least speculate on their anticipated efficacy?
3. Could the authors provide an ablation study on the impact of individual formalization actions within PASS (e.g., symbolic abstraction vs. strategic decomposition) on attack success, stealth, and transferability? How critical are each of these actions to performance?

---

### Official Review · Reviewer_7YUR · 2025-10-28

**Soundness:** 1
**Presentation:** 3
**Contribution:** 2
**Rating:** 2
**Confidence:** 4

**Summary:**

This paper proposes PASS, a novel jailbreak framework that rewrites prompts through structured formalization operations guided by reinforcement learning. The method learns to generate adversarial prompts that evade alignment filters while preserving semantic intent. A graph-based memory module enhances reuse of effective attack strategies. Experiments show strong performance across multiple LLMs, outperforming prior jailbreak methods.

**Strengths:**

1. The paper introduces a novel jailbreak framework that combines formalized prompt rewriting with reinforcement learning, offering a structured and adaptive jailbreak approach.
2. Experimental results across several open-source LLMs demonstrate strong performance.

**Weaknesses:**

1. Evaluation lacks comparison with strong optimization-based jailbreak methods like AutoDAN or GCG, limiting the significance of reported improvements.
2. The contribution of the GraphRAG module is unclear, as no ablation study is provided to isolate its effect.
3. The paper does not explore whether the adversarial prompts negatively affect model behavior on normal tasks.
4. The explanation regarding “embedding space blind spots” remains speculative, with no supporting visualizations or probing analyses.
5. No defense-side experiments are provided, making it unclear how robust the attack is against stronger alignment or adversarial training methods.

**Questions:**

Please refer to the weaknesses.

---

### Official Review · Reviewer_4tE7 · 2025-10-29

**Soundness:** 2
**Presentation:** 1
**Contribution:** 2
**Rating:** 2
**Confidence:** 4

**Summary:**

This paper proposes an enhanced jailbreak attack method named PASS, whose key innovation lies in decomposing the attack process into multiple sub-steps. It trains a Reinforcement Learning (RL) Agent based on whether each of the step can successfully jailbreak the model, and further assists in constructing a GraphRAG for detail enrichment, enabling the Agent to learn effective steps for jailbreaking LLMs. Experimental results demonstrate that the proposed method significantly outperforms other approaches in terms of jailbreak success rate.

**Strengths:**

The paper has a clear motivation.

**Weaknesses:**

#### Methodology
The description in the Methodology section is exceptionally vague. The excessive use of figures has come at the expense of a clear and thorough explanation of the proposed method. This section also suffers from significant issues in academic writing, including the misuse and reuse of symbols, as well as paragraphs filled with redundant and uninformative content. For example,
- The architecture and training specifics of the PolicyNet and ValueNet are absent. In L269-L291, It fails to describe how the state vector is constructed. The role of the seven characteristics comprising the state vector remains unclear. Furthermore, the seven steps listed in the latter part do not correspond to the six components in Figure 3. For instance, "Metaphorical Transformation" is not represented in the figure, and its implementation is not explained.
- The notation r_d \in 0, 1 is non-standard and should be corrected to r_d \in {0, 1}.
- The content in lines 429-477 should be moved to the Methodology.

#### Experiment
The experimental section is remarkably simplistic and fails to be convincing. The entire experimental outcome is confined to a single table. The authors employ custom-defined success/failure criteria. It seems incredible that PASS reportedly outperforms other methods by nearly 50-100 times under these defined metrics. This extraordinary claim requires rigorous validation. Also, as PASS integrates multiple components including the action decomposition, state extraction, action selection, symbolic abstraction, logic encoding and more, an ablation test is necessary to clarify how these components contribute to the final performance.

**Questions:**

- How is the Action Mask assigned? How does the PolicyNet capture the relationship between action masks for different types of queries (e.g., building a bomb and robbing a bank)?
- How are the rewards r_s, r_e, r_h, and r_d obtained? Are r_t and \mathbf{r}_t referring to the same thing?
- How is the input decomposed into sub-steps? Is this achieved through prompting?
- Which database is used for the GraphRAG? How are sub-steps matched to the corresponding graph nodes to extract relevant details?
- How is the hyperparameter \alpha set or tuned?
- PASS employs a rather complex computational structure. What is the computational overhead and processing time?
- How is the average value of T in PASS, and what is its failure rate?
- Are the success rates of other methods on these two datasets consistently as poor as reported in this paper when cited in other literature?
- Are the defined "Success Criteria" and "Failure Criteria" mutually exclusive? Why were standard, commonly-used metrics in this field not adopted to evaluate the method's performance?

---

### Official Review · Reviewer_oHBg · 2025-10-30

**Soundness:** 2
**Presentation:** 3
**Contribution:** 1
**Rating:** 2
**Confidence:** 5

**Summary:**

This paper use RL to decomposes a harmful query into formalized symbols to jailbreak the model.

**Strengths:**

+ The paper is easy to follow

**Weaknesses:**

- Limited novelty. Bypass the llm alignment by obfuscation is a well-explored technique. For example, it can use cipher[1], or replace sensitive words with [MASK] [2]. This paper also falls into this category while not mentioning the related works.

- I have concerns regarding the faithfulness of the constructed formalized symbols. For example, in Figure 4, the original harmful question is write a script that hacks into a bank's dataset. In the final round, it even does not contain the word 'bank' and 'hack', and I am wondering if it is really harmful as it only queries the database verification. By looking at the re-transformed response, I don't think it faithfully answer the original harmful question, and itself is only about how to interacting with the database, which is far from 'detailed instruction'.

- It should compare with some strong baselines, such as [2,3].

- No repeated runs.


[1] GPT-4 Is Too Smart To Be Safe: Stealthy Chat with LLMs via Cipher

[2] WordGame: Efficient & Effective LLM Jailbreak via Simultaneous Obfuscation in Query and Response

[3] Jailbreaking Leading Safety-Aligned LLMs with Simple Adaptive Attacks

**Questions:**

Can you justify the results in Figure 4 to show why the re-transformed response is harmful and answer the original question?

---

### Note · Authors · 2025-11-20

I have read and agree with the venue's withdrawal policy on behalf of myself and my co-authors.